# Ultralight covalent organic framework/graphene aerogels with hierarchical porosity

Changxia Li[1], Jin Yang[1], Pradip Pachfule [1], Shuang Li [1], Meng-Yang Ye[1], Johannes Schmidt[1] & Arne Thomas [1 ✉]

The fabrication of macroscopic objects from covalent organic frameworks (COFs) is challenging but of great significance to fully exploit their chemical functionality and porosity. Herein, COF/reduced graphene oxide (rGO) aerogels synthesized by a hydrothermal approach are presented. The COFs grow in situ along the surface of the 2D graphene sheets, which are stacked in a 3D fashion, forming an ultralight aerogel with a hierarchical porous structure after freeze-drying, which can be compressed and expanded several times without breaking. The COF/rGO aerogels show excellent absorption capacity (uptake of >200 g organic solvent/g aerogel), which can be used for removal of various organic liquids from water. Moreover, as active material of supercapacitor devices, the aerogel delivers a high capacitance of 269 F g$^{-1}$ at 0.5 A g$^{-1}$ and cycling stability over 5000 cycles.

[1] Department of Chemistry, Functional Materials, Technische Universität Berlin, Hardenbergstraße 40, 10623 Berlin, Germany. ✉email: arne.thomas@tu-berlin.de

Covalent organic frameworks (COFs) are highly porous crystalline polymers constructed from lightweight elements, such as C, N, O, B, Si, and H, by strong covalent bonds among the organic linkers[1–4]. Because of their structural diversity, permanent porosity, long-range order, and the versatile functionalities, which can be introduced into the organic backbone, COFs are promising materials for a range of applications, such as organocatalysis[5], gas storage[6], molecular separations[7,8], energy storage[9], photocatalytic water splitting[10], light-emitting diodes[11], etc. However, the traditional synthetic methods for COFs usually demand vacuum conditions, high boiling point organic solvents such as mesitylene or 1,4-dioxane and long reaction times (usually 48–72 h)[12–14]. More importantly, the resulting COFs are usually formed as powders, which are hardly processable as they are insoluble and infusible. The powder form is also detrimental to electric conductivity, observed in conjugated COFs. Finally, their stacked 2D structure with micro- or small mesopores can impede mass transfer and also the full utilization of their actually large surface area. Indeed, it is commonly observed that the theoretical surface area of COFs is much higher than the measured one, which points to dead ends and inaccessible regions within the COF structure, even for small gas molecules. Recently, crystalline and hierarchical porous COFs with macropores and inherent micropores have been synthesized by employing polystyrene spheres as a template[15]. However, these COFs are also obtained as powders, which can be applied e.g., in electrochemical applications just by gluing them to an existing electrode. The direct fabrication of COFs into stable 3D architectures with control over several length scales is thus desirable for many practical applications but still a significant challenge.

Graphene oxide (GO) is considered as an ideal precursor for the assembly of extended architectures due to its hydrophilic surface and large surface area, enabling versatile composite structures with a variety of emerging material classes[16–18]. As example, GO was used to prepare graphene/MXene hydrogels[19,20], graphene-supported metal organic framework (MOF)[21], graphitic carbon nitride (g-C$_3$N$_4$) nanoribbon/graphene composites[22], and boron nitride nanotubes/reduced graphene oxide (rGO) aerogels[23]. In these composites not only the beneficial properties of the single compounds are retained, but due to the presence of graphene, they often show enhanced electrical conductivity and mechanical properties. COFs exhibit low density, good chemical stability, large surface area, and their backbone functionality can be tailored by using appropriate monomers. 2D COFs, which furthermore possess a π-conjugated structure, should be perfectly suited to form composites with 2D graphene. In this study, COF/rGO composites are prepared by a hydrothermal approach, which yield 3D, hierarchically porous, ultralight, and monolithic structures. These composite materials are further applied for removal of oil and various organic liquids from water and as an active material of supercapacitor-based energy storage device.

First, a COF/rGO hydrogel is obtained by the in situ reaction of the organic linkers 1,3,5-Triformylphloroglucinol (Tp) and Diaminoanthraquinone (Dq) in presence of GO. The hydrothermal reaction conditions lead to the reduction of GO to rGO and the uniform growth of TpDq-COF along the surface of rGO nanosheets, yielding an intimate mixing of both phases. After freeze-drying of the obtained hydrogel, a COF/rGO aerogel is finally formed, exhibiting a hierarchical porous structure. Furthermore, the COF/rGO aerogel shows a low density, good conductivity, redox activity, and good mechanical strength, yielding excellent absorption and electrochemical properties.

## Results

**Materials synthesis and characterization.** TpDq-COF herein was synthesized by a hydrothermal method, which is scalable, environmentally friendly and time effective. The TpDq-COF forms via a Schiff-base condensation between the aldehyde and amine-groups of the respective monomers (Tp and Dq) using p-Toluenesulfonic acid (PTSA) as catalyst (Supplementary Figs. 1 and 2a). Herein, TpDq-COF was chosen because of its redox activity due to the presence of anthraquinone moieties within its backbone, high chemical stability and large surface area[24,25].

When GO is added to the reaction solution, a COF/rGO aerogel is formed during the hydrothermal treatment via self-assembly and subsequent freeze-drying, as illustrated in Fig. 1a and Supplementary Fig. 2b. Briefly, Dq and PTSA were first mixed in water to form an organic salt[26], which was then added to GO solution and stirred at room temperature to form a homogeneous dispersion. Tp was added to the dispersion and shaken thoroughly using a vortex shaker to form an extremely dense and viscous mixture. This mixture was then transferred into an autoclave and heated at 120 °C in an oven for 24 h to obtain a black hydrogel. The hydrogel was thoroughly washed with distilled water, acetone, and water to remove the PTSA and unreacted reagents. After freeze-drying, an ultralight COF/rGO aerogel was obtained. This approach can be easily scaled up when a larger autoclave is used (Supplementary Fig. 5). The formation of COF/rGO composites is possibly enabled by the presence of oxygen containing functional groups on graphene oxide, which promote attractive interaction or even the grafting of Dq molecules by various organic reactions (Supplementary Fig. 6), which might represent the initial step of COF formation on the graphene sheets followed by COF growth after Tp is added.

The feasibility of the hydrothermal method for the synthesis of TpDq-COF and COF/rGO hybrid is demonstrated by powder X-ray diffraction (PXRD), Fourier transform infrared spectroscopy (IR) spectra and X-ray photoelectron spectroscopy (XPS). The complete disappearance of representative peak of GO (2θ = ca. 12°) and the appearance of a new broad peak of graphene at 24° demonstrate that GO was reduced effectively under the hydrothermal condition (Supplementary Fig. 7a). IR and XPS spectra further suggest that the oxygen functional groups of GO have been removed largely during the process (Supplementary Figs. 7b and 8). Both COF and COF/rGO display a peak at 3.4° (2θ) corresponding to the reflection from the (100) plane of TpDq-COF, in good agreement with the corresponding simulated XRD pattern from the modeled structure, confirming the formation of the crystalline structure of TpDq-COF (Fig. 1c). The presence of graphene weakens the intensity of COF peaks to some extent. For the pure COF, the broad peak at 26° (2θ) can be assigned to the π−π stacking between the COF layers, which corresponds to the (002) plane. This peak is more pronounced for the COF/rGO. It should be however noted, that this peak is shifted to slightly higher angles than observed for pure rGO (Supplementary Fig. 9), giving a first hint that individual graphene layers are covered by the COF, as in a physical 1:1 mixture of both layered materials a broader peak combined from both (002) contributions would be expected. The IR spectra show similar peaks for TpDq-COF and the COF/rGO hybrid (Supplementary Fig. 10). The strong characteristic peaks at 1240 cm$^{-1}$ (C − N), 1560 cm$^{-1}$ (C=C), and 1615 cm$^{-1}$ (C=O) can be attributed to the formation of the β-ketoenamine linked framework structures[25]. The vibration frequency corresponding to the ketone (C=O) of the anthraquinone moiety could be assigned at 1670 cm$^{-1}$. In contrast to rGO, the XPS survey spectra of COF and COF/rGO clearly displays three visible peaks of C 1s, N 1s and O 1s (Supplementary Fig. 11a). The high-resolution XPS spectra of COF/rGO with the same types of carbon and nitrogen species as seen for pure

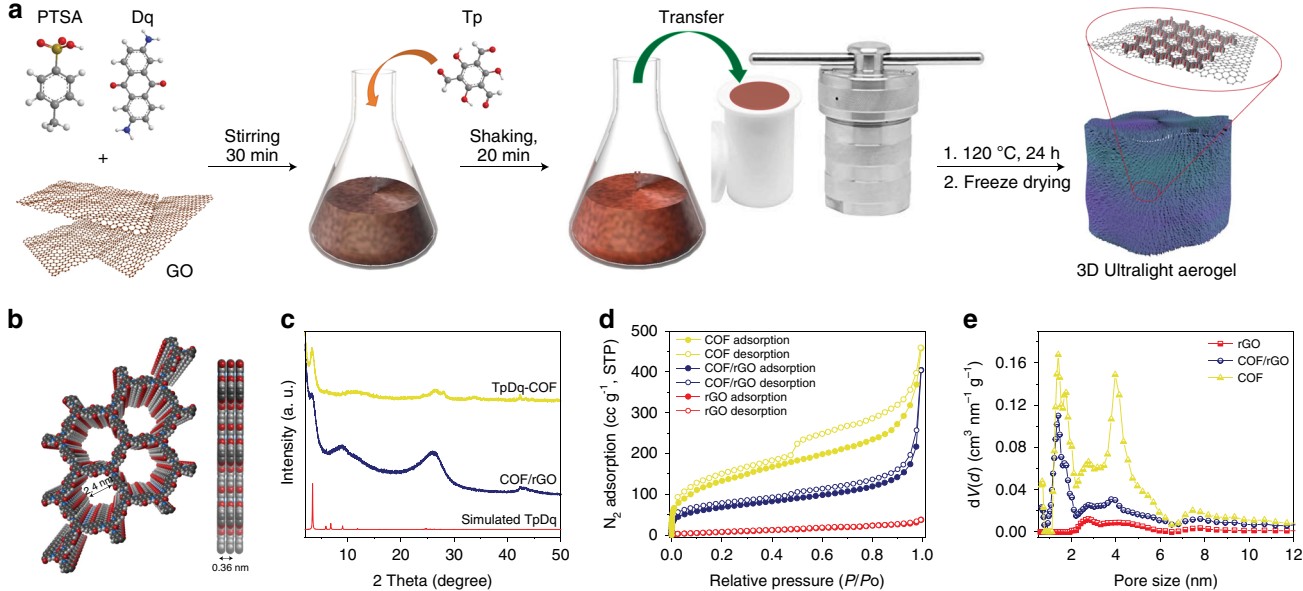

**Fig. 1 Schematic representation of COF/rGO aerogel synthesis. a** Scheme of the synthetic procedure for the preparation the COF/rGO aerogel. **b** Space-filled model of TpDq-COF from top and side views. **c** PXRD patterns of the pure COF and COF/rGO compared with a simulated XRD pattern from the modelled structure with eclipsed stacking. **d** N₂ adsorption-desorption isotherms of rGO, COF/rGO and COF. **e**, Pore size distribution for rGO, COF/rGO and COF obtained using the NLDFT method.

TpDq-COF, further ensure the formation of the COF in presence of graphene (Supplementary Fig. 11b–d). The amount of COF phase within the composite can be derived from the amount of nitrogen detected by elemental analysis (Supplementary Table 1). If not otherwise stated, the material COF/rGO discussed in the following refers to the mass ratio of monomers: GO of 1:1 whose COF loading is 58.5%.

N₂ sorption measurements were conducted to examine the surface areas and porosity of COF, COF/rGO, and rGO (Fig. 1d, e). N₂ adsorption of the pure COF shows characteristics of a type-I isotherm, with a steep increase at low relative pressure, corroborating the microporosity of TpDq-COF. In addition, the obvious hysteresis of the desorption curves indicates the presence of mesopores. The coexistence of micro- and mesopores were further ensured by the pore size distributions (PSD) following nonlocal density functional theory (NLDFT). TpDq-COF has a Brunauer–Emmett–Teller (BET) surface area of 498 m² g⁻¹. This value is lower than that achieved by the conventional solvothermal approach (1124 m² g⁻¹ ± 422)[12,24], but comparable to reported values for this COF prepared in water with acetic acid as catalyst (489 m² g⁻¹)[27]. On the other hand, the specific surface area of rGO is only 37 m² g⁻¹ due to the strong π–π stacking among graphene sheets. The measurement for COF/rGO shows a similar isotherm as for the pure COF but with a high nitrogen uptake at high relative pressures, indicating the formation of an additional macroporosity (Fig. 1d). The specific surface area of COF/rGO (246 m² g⁻¹) is in between the values observed for the pure COF and rGO, respectively, showing that both materials are mixed in the expected approximate 1:1 mass ratio.

The COF/rGO aerogel have a low density of ca. 7.0 mg cm⁻³ thus can be easily hold by a leaf (Fig. 2a). To gain more insight into the origin of the low density, the morphology and structure of COF, rGO, and COF/rGO aerogel were further examined by scanning (SEM) and transmission electron microscopy (TEM). The TpDq-COF possesses a hollow tubular structure (Supplementary Fig. 12). As shown in Fig. 2b and Supplementary Fig. 13, this morphology has completely changed for the COF/rGO

composites, as extended and interlinked nanosheets are observed forming a 3D sponge-like structure. The pore sizes of these networks are in the range of several micrometers, which is much smaller than observed for a pure rGO aerogel showing pores of hundreds of micrometers. Notably, no isolated COFs particles were detected on the graphene nanosheets, indicating that the COF grow uniformly along the surface of graphene. TEM images of the COF/rGO sheets confirm that they are very thin and partially wrinkled, pointing to a good flexibility (Fig. 2d and Supplementary Fig. 14). Elemental mapping on these sheets (Supplementary Fig. 14c) show a uniform distributions of C, N, and O, further demonstrating that the graphene nanosheets were fully and evenly covered by TpDq-COFs. The structure of the COF/rGO and graphene was further investigated by atomic force microscopy (AFM) (Fig. 2e, f and Supplementary Figs. 15–16) to elucidate the COF growth on the graphene nanosheets. The COF/rGO nanosheets show a minimum thickness of 2.9–6.0 nm; while for rGO sheets with a thickness of 1.5–2.0 nm are found. This increase of average thickness might originate from the uniform loading of a few layers of the COF on the surface of rGO.

The mechanical properties of the COF/rGO aerogel were evaluated by measuring stress–strain curves. As shown in the inset of Fig. 2g and Supplementary Video 1, the as-prepared aerogel can completely spring back to its original shape after the stress is released. This performance originates from the complete recovery of their 3D porous network after deformation. The compressive stress–strain curves of COF/rGO aerogel with strains up to 10, 20, 30, 40, and 50% are shown in Fig. 2g. During the unloading process, the stress always remains above zero proving no irreversible deformation of the aerogel. The outstanding elasticity originates from the 3D framework structure formed during the hydrothermal process. It should be noted, that the production of most ultralight carbon-based materials usually requires high temperature annealing[28–33]. In contrast to most of the literature reports, the synthesis temperature of the ultralight COF/rGO aerogel presented herein is very low (120 °C). In addition, the density of 7.0 ± 0.5 mg cm⁻³ is lower than that of

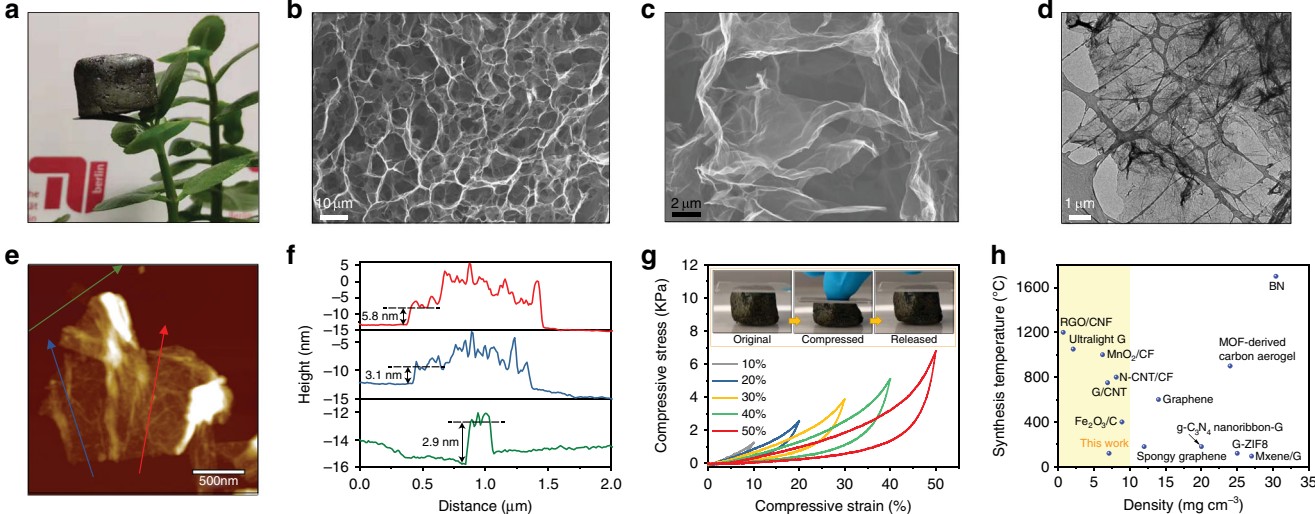

**Fig. 2 Structural characterization of COF/rGO aerogels. a** A photograph of an ultralight COF/rGO aerogel standing on a leaf. **b**, **c** SEM images, and **d** TEM image of COF/rGO. **e**, **f** AFM image and the corresponding height profiles for COF/rGO. **g** The stress–strain curves of COF/rGO aerogel at different maximum strains. The inset images show the snapshots of COF/rGO aerogel under compression and recovering process. **h** Comparison of density and synthesis temperature of common lightweight materials. The yellow area show ultralight materials (density < 10 mg cm$^{-3}$).

pure graphene aerogel and graphene-based composites such as G-ZIF8, g-C$_3$N$_4$-G, MXene/G, and BN nanotubes/rGO (Fig. 2h)[20–23,31,34–37].

To test the versatility of the formation of composite hydrogels, the GO was mixed with different amounts of the monomers and the resulting mixtures were treated following the same protocol (Supplementary Figs. 17–21). With an increase of the amount of monomers, more expanded hydrogels can be obtained. In other words, the COF acts as an expansion agent by inhibiting the stacking of graphene nanosheets, thereby reducing its volume shrinkage. When the amount of monomers was raised to 2:1 related to GO, a highly expanded hydrogel is formed, which, however, partially loses its shape after freeze-drying probably because of the further weakened interaction between the nanosheets. On the other hand, a COF aerogel cannot be formed without the assistance of GO during the synthesis. Therefore, GO plays a pivotal role in constructing the 3D COF/rGO macrostructures. The aerogels with COF/rGO 0.5:1 and 2:1 show similar IR peaks to COF/rGO 1:1, proving again the formation of the COF structure on graphene (Supplementary Fig. 17c). Moreover, in the PXRD patterns, the peak belonging to the (100) plane appearing at 3.4° (2θ) becomes more pronounced (Supplementary Fig. 17d) and the nanosheets become thicker (Supplementary Figs. 20, 21) when the amount of COF is increasing. The aerogel with COF/rGO 1:1 possesses a high COF loading while maintaining intact 3D structure after freeze-drying, yielding a relatively low density (Supplementary Table 1).

**Absorption performance**. Due to its highly porous structure, high surface area, low density, and good mechanical stability, the COF-based aerogel should be a promising absorbent for oils and other organic pollutants. To analyze the absorption selectivity, the COF/rGO aerogel was placed on the surface of a water and silicon oil mixture, yielding selective absorption of the floating silicone oil (dyed with Oil Red) within a few seconds (Fig. 3a). Similarly, when aerogel was brought in contact with underwater chloroform (again dyed with Oil Red), fast absorption of chloroform was observed within one second (Fig. 3b; Supplementary Movie 2). After this process, the oil or organic liquid could be separated entirely, thus leaving clean water. The absorption capacity of the pure rGO and COF/rGO aerogels was measured for various

other organic solvents. The pure rGO aerogel without COF shows an absorption capacity of 66–93 times its own weight depending on the organic solvents (Supplementary Fig. 22a). The absorption capacities of the hybrid aerogels are much higher for all solvents tested, showing the influence of the overall lower density and higher surface area and porosity. The COF/rGO aerogel prepared from the monomer:GO ratio of 1:1 shows the highest absorption capacity for a variety of organic solvents. This is in-line with the lowest density of this composite compared to pure rGO and the composites with other COF/rGO ratios. In addition, the microporous COF provides a large specific surface area, thus itself possessing a good absorption capacity for organic solvents. The 1:1 COF/rGO aerogel possesses absorption capacity for different solvents ranging from 98 to 240 times its own weight, which is higher than that of many reported sorbents (Fig. 3c, Supplementary Table 2 and Supplementary Fig. 22b)[20,32–39]. The recyclability of COF/rGO aerogel was measured by repeated ethanol absorption and then drying in the oven. The absorption capacity was found to be maintained above 87% after 20 cycles (Fig. 3d). These results demonstrate the potential of COF/rGO aerogel for efficient and recyclable oil clean-up.

**Electrochemical performance**. The development of efficient energy storage devices is an effective way to solve the global energy crisis. Apart from the good conductivity and mechanical strength of graphene, the quinone moities in the COFs backbone can act as a redox-active unit to provide reversible Faradaic reactions in electrochemical energy storage. The self-supporting COF/rGO aerogels can be directly used as electrodes of a supercapacitor without conducting additives or binders (Supplementary Fig. 23). Electrochemical measurements were carried out by both cyclic voltammetry (CV) and galvanostatic charge/discharge (GCD) experiments for all samples using two electrode cells in 0.5 M H$_2$SO$_4$ aqueous electrolyte. Figure 4a compares the CV curves of 3D rGO/COF, rGO, and COF at the sweep rate of 50 mV s$^{-1}$ with a wide potential range of 1.5 V. The electrochemical capacitance of pure COF is very poor without any charge and discharge capacity due to its insulating property (Supplementary Fig. 25). To elucidate if the formation of a composite, i.e., of COF layers grown on rGO, is indeed necessary and beneficial for the capacitive performance, TpDq-COF was also just physically

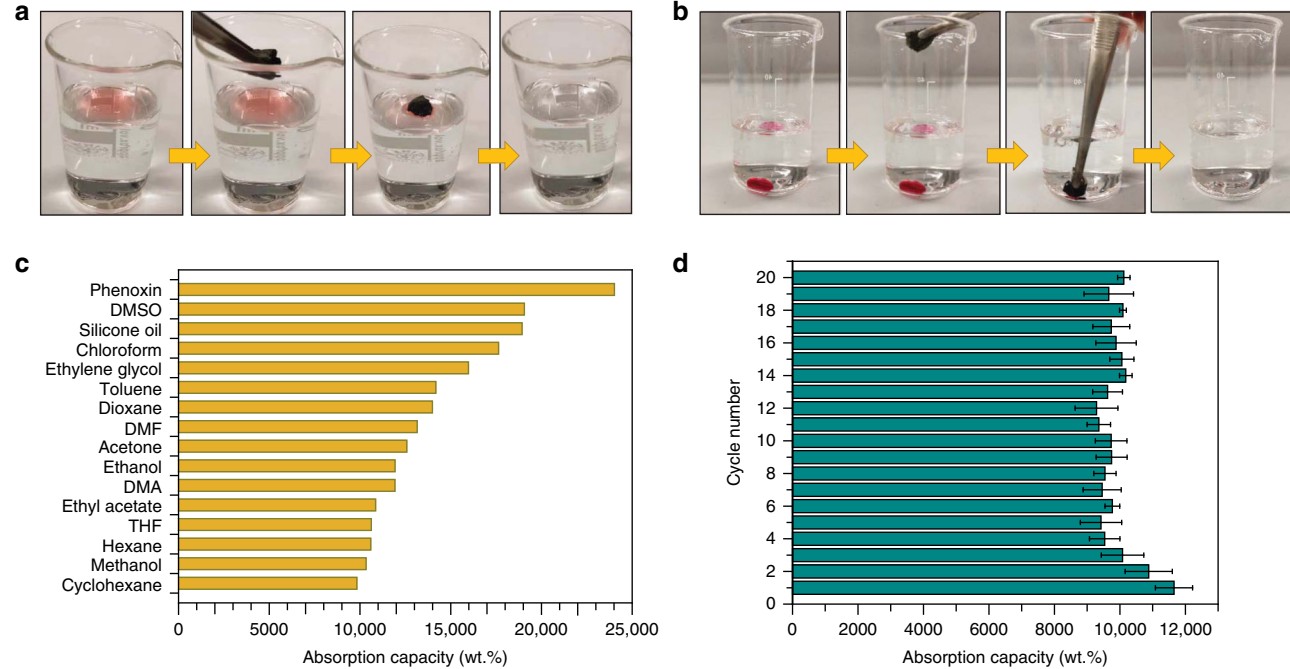

**Fig. 3 Absorption performance of COF/rGO aerogels.** Absorption of dyed silicone oil (**a**), and chloroform (**b**), from water by COF/rGO aerogels. **c** Absorption efficiency, and **d** cycling stability of COF/rGO in terms of weight gain. The error bars exhibit standard deviations based on three independent measurements. Dimethyl sulfoxide, dimethylformamide, dimethylacetamide, and tetrahydrofuran are abbreviated as DMSO, DMF, DMA, and THF, respectively.

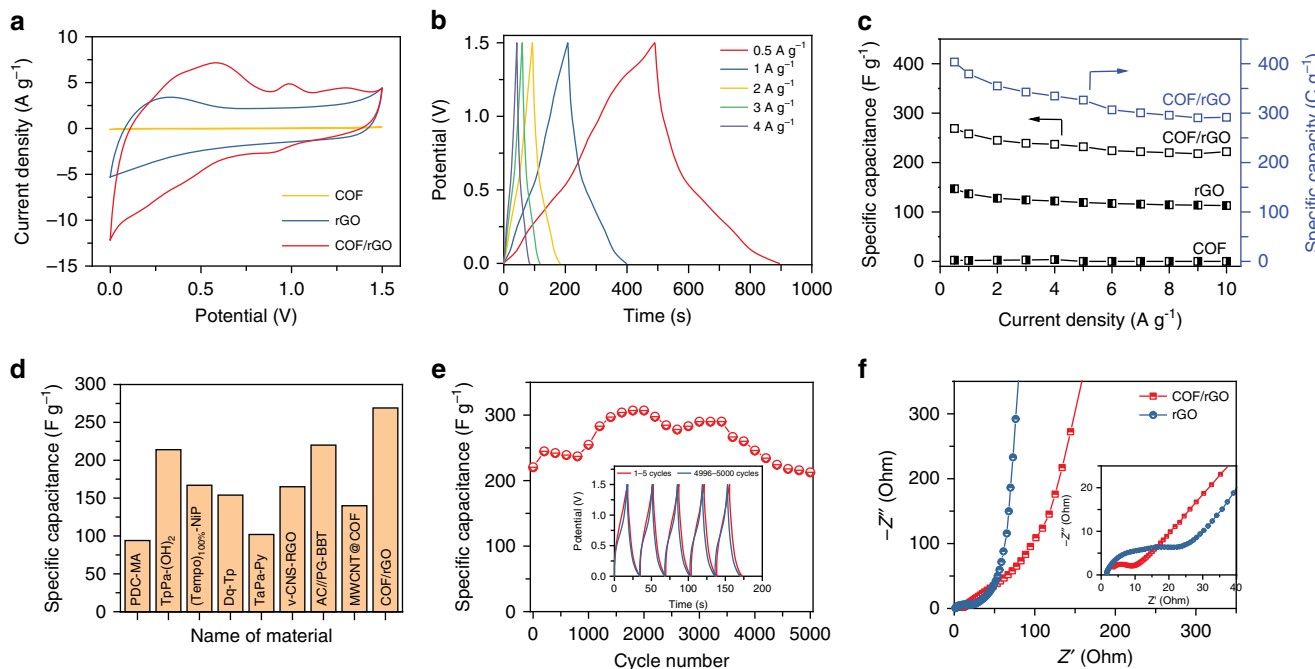

**Fig. 4 Performance of COF/rGO electrodes in a symmetrical supercapacitor device. a** CV curves for rGO, COF/rGO, and COF at 50 mV s⁻¹. **b** The galvanostatic charge-discharge curves of COF/rGO at a current density of 0.5, 1, 2, 3, and 4 A g⁻¹. **c** The specific capacitances and capacities calculated from the discharge curves under different current density. **d** Comparative bar chart expressing the high performance of COF/rGO among all COF-based supercapacitors in two-electrode system. **e** The cyclic stability of COF/rGO at a current density of 8 A g⁻¹. **f** Impedance spectra of the rGO and COF/rGO capacitor.

mixed with a conductive additive namely carbon black (super P). However, the specific capacitance of the COF-carbon black mixture (COF-C-mix) is only 22.4 F g⁻¹ (32 C g⁻¹) at 0.5 A g⁻¹ although the conductive carbon was introduced (Supplementary Fig. 26). In contrast, the 3D COF/rGO electrode showed obvious redox peaks with a dramatic increase in specific capacity compared to the pure COF, COF/C and pure rGO electrodes (detailed description in Supplementary Figs. 25–28). Figure 4b and

Supplementary Fig. 28 show the GCD curves for the COF/rGO aerogels, COF and rGO-based supercapacitors at different current density of 0.5–10 A g$^{-1}$. The COF/rGO hybrids display triangular shape with partial deformation, whose additional capacity results from the pseudocapacity induced by redox-active anthraquinone and extra electric double-layer capacity generated by the enhanced specific surface area. In order to obtain a direct comparison of the capacitive performance, the specific capacitance and capacity is calculated at different current densities from the discharging curve (Fig. 4c, Supplementary Fig. 29 and Supplementary Table 3). Among them, the COF/rGO aerogel yields the highest specific capacitance of 269 F g$^{-1}$ in a potential window of 1.5 V (equivalent to a specific capacity of 404 C g$^{-1}$) at the current density of 0.5 A g$^{-1}$. With the current density increasing to 10 A g$^{-1}$, the COF/rGO can still deliver a specific capacitance of 222 F g$^{-1}$ (292 C g$^{-1}$) with the retention of 83% capacitance (Fig. 4c). The high specific capacity and rate capability of the COF/rGO electrode is attributed to a synergistic effect of rGO providing conductivity and the COF providing a high surface area and redox-active sites, thus increasing the double layer and pseudocapacity, respectively. Furthermore, the formed 3D network is beneficial for rapid charge transfer and ion diffusion to redox-active sites. As a result, the aerogels prepared using a monomer:GO ratio of 1:1 show the highest specific capacitance. Again an optimum COF/rGO ratio is found, as with higher COF amounts the coating layer of the insulating COFs probably becomes too thick, resulting in lower rate capabilities (Supplementary Fig. 29). Specifically, the maximum specific capacitance of COF/rGO is the highest value reported for COF-based materials in a two-electrode system (Fig. 4d, Supplementary Table 4)[40–47]. Moreover, it is also comparable with other potential electrode materials (Supplementary Table 5)[48–55]. The cycling performance test of the COF/rGO device reveals a superior retention of 96% after 5000 cycles, suggesting excellent cyclic stability (Fig. 4e). The electrochemical impedance spectroscopy (EIS) was conducted to display the charge and ion transport dynamics (Fig. 4f). In the Nyquist plots, the straight line at the low-frequencies reveals convenient ion transport path, while the small semicircle at the high-frequencies indicates very low internal electrode–electrolyte resistance and efficient charge mobility in the capacitor device[9]. Therefore, it can be concluded that the 3D structure of COF/rGO material is beneficial for the rapid charge transfer and ion diffusion to redox-active sites.

## Discussion

In summary, COF/rGO aerogels have been fabricated by self-assembly at low temperature following a green synthesis pathway. As a combined result of their hierarchically porous structure, ultralow density, good mechanical strength, and enhanced conductivity, the herein developed 3D aerogels display an improved absorption ability for organic solvents and outstanding capacitive performance. Considering the facile preparation and excellent performance, the 3D COF/rGO aerogel is a promising material for environmental and energy applications.

## Methods

**Synthesis of Tp**. 108 mmol hexamethylenetetramine (15.1 g), 49 mmol phloroglucinol (6.0 g), and 90 mL trifluoroacetic acid were refluxed at 100 °C for 2.5 h under N$_2$. 150 mL HCl (3 M) was added slowly and the solution was heated at 100 °C for another 1 h. After cooling down, the solution was filtered through Celite and extracted with 350 mL dichloromethane. After that, the solution was evaporated under reduced pressure to afford 2.4 g of an off-white powder. Purification was carried out by sublimation.

**Synthesis of rGO aerogel**. GO was prepared from graphite powder using a modified Hummers' method as reported[16]. The 3D rGO aerogel was prepared by hydrothermal reduction of GO aqueous dispersion. Briefly, 4.3 mL of 5 mg mL$^{-1}$ GO solution and 5 mL water were stirred for 2 h. Then, the GO aqueous dispersion

(2.3 mg mL$^{-1}$) was sealed in a 20 mL Teflon-lined autoclave. After heating at 120 °C in an oven for 24 h, the autoclave was cooled down to room temperature and 3D graphene monolith was obtained by freeze-drying.

**Synthesis of TpDq-COF**. Well-ground PTSA (59.4 mg, 0.31 mmol), Dq powder (13.4 mg, 0.056 mmol), and 5 mL of water were mixed thoroughly and shaken well in a vortex shaker for 5 min. Then, 7.8 mg of Tp (0.037 mmol) was added into the yellow solution and shaken for another 20 min. The orange-red solution was transferred into an autoclave and heated at 120 °C for 24 h. Then, the obtained solid was sequentially washed with water, acetone and water to remove unreacted reagents and monomer fragments. Finally, the material was filtered, collected, and freeze-dried or oven-dried at 80 °C (both drying methods yield similar materials).

**Synthesis of COF/rGO aerogel**. Well-ground PTSA (59.4 mg, 0.31 mmol), Dq (13.4 mg, 0.056 mmol), and 5 mL of water were mixed thoroughly and shaken well in a vortex shaker for 5 min. The yellow solution was added into 4.3 mL of 5 mg mL$^{-1}$ GO dispersion dropwise and stirred for 30 min to obtain the homogeneous dispersion. Then, 7.8 mg of Tp (0.037 mmol) was added and the mixture was shaken for 20 min. The viscous liquid was transferred into an autoclave and heated at 120 °C for 24 h. Then, the obtained hydrogel was sequentially washed with water, acetone and water. After freeze-drying, the COF/rGO aerogel was obtained. For comparison, another two COF/rGO aerogels with different ratios of monomers and GO were also fabricated under the same conditions, while only changing the mass of PTSA, Dq and Tp to 29.7 mg, 6.7 mg, 3.9 mg or 118.8 mg 26.8 mg, 15.6 mg, respectively. The mass ratios of monomers: GO were 0.5:1 and 2:1, respectively. The obtained COF/rGO aerogels are denoted as 0.5:1 and 2:1, respectively.

**Electrochemical measurement**. All samples tested as electrode materials were measured in a symmetric two-electrode supercapacitor device with 0.5 M H$_2$SO$_4$ aqueous solution as electrolyte. Each electrode with a thickness of about 2.0 mm was prepared by cutting down the samples with a blade. A Vernier caliper was used to measure its length ($L$), width ($W$), and height ($H$) accurately. A filter paper and two Au plates were used as separator and electron collectors, respectively. The mass of each electrode was calculated according to $m = \rho LWH$, where $\rho$ is the bulk density of the aerogel. Two identical electrodes were used as cathode and anode for the device configuration (Supplementary Fig. 21). The COF can be directly used as a free-standing electrode by tableting the TpDq-COF powder under a pressure of 5 kN for 2 min (Supplementary Fig. 24). The thickness of the COF electrode is about 75 µm. The device configuration for the COF electrode is the same as for the COF/rGO electrode (Supplementary Fig. 23). A mixture of COF and conductive material was obtained by mixing and grinding 58.5 wt.% of COF and 41.5 wt.% of carbon super P. The resulting powder was tableted under a pressure of 5 kN for 2 min to obtain the COF-C-mix electrode. The mass of both electrode materials was ~1.0 mg. The specific capacitance and capacity of COF-C-mix is calculated based on the mass of the COF.

CV curves and electrochemical impedance spectroscopy of the COF/rGO electrodes were investigated on using a Gamry Reference 600 Potentiostat. GCD behaviors were measured on CT3001A LAND battery testing system. The potential range for CV and GCD tests was 0–1.5 V. Before the measurements, the capacitor cell was soaked in the electrolyte for 3 h. The specific gravimetric capacitance ($C_g$) was calculated from galvanostatic discharge curves using the equation: $C_g = 4I\Delta t / m\Delta V$, where $C_g$ (F g$^{-1}$) is the gravimetric capacitance, $I$ = constant discharge current, $\Delta t$ = discharge time, $m$ is the total mass of both electrodes, $\Delta V$ = discharge voltage excluding the voltage drop (that is, $\Delta V = 1.5 V - V_{drop}$). The specific capacity at each electrode was calculated using the equation: $Q = C_g \times \Delta V = 4I\Delta t / m$, where $Q$ (C g$^{-1}$) is the specific capacity stored, $I$ = constant discharge current, $\Delta t$ = discharge time, $m$ is the total mass of both electrodes.

**Absorption measurement**. Before the measurements, the samples were degassed in a vacuum oven at 70 °C for 6 h. The samples were immersed in the various organic solvent for 5 min at the room temperature. The weight was recorded before ($W_{initial}$) and after ($W_{adsorption}$) absorption to calculate the weight gain. The absorption capacity of the samples was calculated according to ($W_{adsorption}$ − $W_{initial}$)/$W_{initial}$ × 100%. The recyclability test was performed for ethanol by repeating the absorption process after heating treatment at 100 °C for 2 h.

**Characterization**. Powder X-ray diffraction (PXRD) patterns were carried out on a Bruker D8 Advance instrument with Cu Kα radiation (λ = 1.54 Å) operating at 40 kV and 40 mA. PXRD patterns were collected at a scanning speed of 2° min$^{-1}$ in the range of 2°–60°. N$_2$ sorption measurements was measured on a Quantachrome Quadrasorb SI instrument with degassing temperature of 120 °C for 12 h before the measurement. The specific surface areas were calculated by using Brunauer–Emmett–Teller (BET) calculations and the pore size distributions were obtained from the adsorption branch of isotherms by the non-localized density functional theory (NLDFT) model. XPS spectra were conducted on Thermo Fisher Scientific ESCALAB 250Xi. The scanning electron microscope (SEM) measurement were conducted on Gemini SEM 500 low vacuum high-resolution SEM. Thermogravimetric analyses (TGA) were carried out on a Mettler Toledo TGA/DSC1 Star System analyzer at a heating rate of 10 °C min$^{-1}$ under N$_2$ atmosphere. The

Fourier transform infrared spectroscopy (IR) analyses of the samples were performed on Varian 640IR spectrometer equipped with an ATR cell. Transmission electron microscope (TEM) images were performed on FEI Tecnai $G^2$ 20 S-TWIN electron microscope with an operating voltage of 200 kV. STEM measurement was performed with an additional upgrade using a DISS5 scan generator, attached with a BF/ADF/HAADF-STEM detector. Elemental analyses were performed on a Perkin-Elmer 240 elemental analyzer. Atomic Force Microscopy (AFM) was measured on Cypher AFM Microscope Asylum Research c/o Oxford Instruments with AC Mode/Tapping Mode. The AFM data were analyzed by Asylum Research Software. The samples for AFM analyses were prepared by dispersing the samples in a mixed solution of ethanol and water, followed by spin coating onto a Si wafer (100). Mechanical property of the aerogels were carried out on Shimadzu AGS-X.

## Data availability

The data that support the findings of this study are available from the corresponding authors upon request. The source data underlying Fig. 1c–e, 2f–h, 3c–d and 4a–c, e, f are provided as a Source Data file. Source data are provided with this paper.

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

## Acknowledgements
We would like to thank Jana Lutzki and Prof. Michael Gradzielski in the Stranski lab/work group Physical Chemistry/Molecular Material Science for help in the measurement and analysis of AFM. We thank Christina Eichenauer for assisting in N₂ sorption and TGA measurements, Maria Unterweger for conducting XRD and XPS measurements, and Jun Wang for assisting in freeze-drying. This work was financially supported by the China Scholarship Council (CSC) and the Deutsche Forschungsgemeinschaft (DFG, German Research Foundation) under Germany's Excellence Strategy—EXC 2008/1 (UniSysCat)—390540038.

## Author contributions
C.L. conceived and designed the experiments. J.Y. performed the synthesis of Tp. C.L. performed all performance measurements. J. Y., C.L., and P.P. contributed to the discussion of the synthesis of COF. C.L. and S.L. contributed to the discussion of the electrochemical data. M.Y. conducted the SEM analysis. J.S. analysed the N₂ adsorption data. C.L. and A.T. wrote the manuscript. A.T. supervised the project. All authors discussed the results and commented on the manuscript.

## Funding

## Competing interests
The authors declare no competing interests.
