## [Peer Review File · Nature Communications]

REVIEWER COMMENTS

Reviewer #1 (Remarks to the Author):

In this manuscript, Li et al reported a facile and pyrolysis-free method for synthesis of COF/reduced graphene oxide (rGO) aerogels. This aerogel was demonstrated to show good absorption capacity and high electrochemical capacitance. This synthetic method show great potential for green construction of COF-based materials, especially the using of water as the reaction media. The paper is well organized, however, some points should be further clarified. A major revision is needed for further consideration.

1. Is there a possibility of oxygen functional groups of graphene oxide participating in the initial polymerization of COF? Why the positive shift of XRD peak can prove the coverage of COF on rGO layer (line 118~120, page 5)?
2. No evidence in Fig. 1d and 1ec could be found to prove the formation of an additional macroporosity in COF/rGO.
3. The rGO aerogel with such low BET surface area ($37 \text{ m}^2 \text{ g}^{-1}$) was demonstrated to show an absorption capacity of 66-93 times, which indicated that these macropores played the vital role for the absorption of organic solvents. As such, why the incorporation of microporous COF could enhance the absorption capacity? And why did the sample of monomers:GO ratio of 1:1 show the highest absorption capacity? Similar concern also exists in their electrochemical properties. Why the insulated COF component could boost the capacitance of rGO aerogel?
4. The fabrication method of electrode for pristine COF powder should be added in the experiment part.

Reviewer #2 (Remarks to the Author):

In this manuscript, the authors described an interesting method for the preparation of 3D hierarchical porous covalent organic frameworks (COFs) beyond the traditional powdery COFs. The COFs are grown in situ along the surface of reduced graphene oxide (rGO) to form a hybrid aerogel (COF/rGO), which exhibits impressive absorption capacity towards organic solvents and delivers as well superior electrochemical performance as electrodes of supercapacitors. The strategy demonstrated here enriches the COF-based materials and facilitates the potential applications of COFs. Thus, this paper is acceptable for publication in Nature Communications, however the following issues should be addressed carefully.

1. Page 9, line 235-239: "The pure rGO aerogel without COF shows an absorption capacity of 66–93 times its own weight depending on the organic solvents (Supplementary Fig. 19a). The absorption capacities of the hybrid aerogels are much higher, showing the influence of the overall lower density, higher surface area and porosity." The explanation is not so convincing. As we know, graphene aerogels with huge absorption capacity can be obtained as well by simply controlling the architectures, and actually various graphene aerogels have been reported. So, the role of the COF in the COFs/rGO hybrids should be further supported and discussed.

2. COFs/rGO hybrids are described as pseudocapacitive electrode materials, and their electrochemical properties in SCs are evaluated with specific capacitance which maybe inappropriate although many literatures have described their electrode materials exhibiting similar behaviors in the same way. The “pseudocapacitance” is assigned to the electrode materials which exhibit a linear dependence of the charge stored with the width of the potential window. While, the specific capacitance of COFs/rGO are “average” values calculated over a limited potential window, which is not constant. As a result, the capacity in C g⁻¹ or mAh g⁻¹ should be chosen to evaluate the electrochemical performance instead of the specific capacitance. Detailed information about this issue is available in these literatures (“To Be or Not To Be Pseudocapacitive?”, *Journal of The Electrochemical Society*, 162 (5) A5185-A5189 (2015); “Redox-Enhanced Electrochemical Capacitors: Status, Opportunity, and Best Practices for Performance Evaluation”, *ACS Energy Lett.* 2017, 2, 11, 2581-2590) or others.

3. Page 11, line 269-271: “The electrochemical capacitance of pure COF is very poor without any charge and discharge capacity due to its insulation property 271 (Supplementary Fig. 22).” The pure COF was then used as the electrodes directly and was compared to the COF/rGO electrode. This comparison is somehow “unfair” because pure COF is insulated and thus does not deliver electrochemical performance. It is maybe more reasonable that pure COF and equivalent conductive materials, such as rGO, should be chosen to fabricate the electrodes and compare to the COF/rGO. Page 11, line 275-278: “The COF/rGO hybrids display triangular shape with partial deformation, whose additional capacitance result from the pseudocapacitance effects induced by redox active anthraquinone”. It may not appropriate that the authors ascribe the additional capacitance only to the pseudocapacitance. The specific surface area of COF/rGO (246 m² g⁻¹) is higher than that of rGO (37 m² g⁻¹). That means COF/rGO should possess higher electric double-layer capacitance compared to rGO. Thus, the additional performance should include pseudocapacity and extra electric double-layer capacity.

Reviewer #3 (Remarks to the Author):

In this manuscript, the authors reported the fabrication of COF/reduced graphene oxide (rGO) aerogels, which show good absorption capacity for organic solvents and can be used as active material of supercapacitor devices with high performance. This is an interesting result, as they provide a new way to fabricate COFs into stable 3D architectures that may have practical applications in many areas. I would like to recommend the publication in *Nature Communications* after addressing the following concerns:

1. As mentioned in the introduction part, GO was used to prepare many different composites, such as graphene-supported MOFs. The author should discuss more here about the advantages of using COF. In addition, the authors can compare the performance of COF/GO composite with other reported systems in the maintext.

2. The author chose TpDq-COF to demonstrate their idea. How about the universality of this method? Is it possible to construct with other other COFs systems, such as some highly porous 3D COFs (*Angew. Chem. Int. Ed.*, 2019, 58, 9770)?

3. According to the literature (J. Am. Chem. Soc. 2013, 135, 16821), TpDq-COF has a pore size at 2.0 A. However, Figure 1e indicated TpDq-COF has two different pores. Why? The author can give some explanation here.

4. The authors claimed the TpDq-COF grow uniformly along the surface. Please give some direct evidences here.

5. Some related literatures about COFs with new applications (Nat. Commun., 2018, 9, 5234; Nat. Commun.,2019, 10, 2467; etc) can be added in the revised manuscript.

Response to reviewers' comments for NCOMMS-20-12679

Reviewer #1 (Remarks to the Author):

In this manuscript, Li et al reported a facile and pyrolysis-free method for synthesis of COF/reduced graphene oxide (rGO) aerogels. This aerogel was demonstrated to show good absorption capacity and high electrochemical capacitance. This synthetic method show great potential for green construction of COF-based materials, especially the using of water as the reaction media. The paper is well organized, however, some points should be further clarified. A major revision is needed for further consideration.

1. Is there a possibility of oxygen functional groups of graphene oxide participating in the initial polymerization of COF? Why the positive shift of XRD peak can prove the coverage of COF on rGO layer (line 118~120, page 5)?

Response: We thank the reviewer for his valuable comments. Indeed, we also believe that oxygen functional groups, such as $-\text{COOH}$ or $-\text{C}=\text{O}$, on graphene oxide at least promote the interaction of the COF monomers and growing COF oligomers to the sheets by hydrogen bonding/coulombic interactions. They might even allow covalent binding of the monomers, such as the aminoanthraquinone molecules, onto the sheets by various organic reactions (*Phys. Chem. Chem. Phys.*, 2011, **13**, 11193–11198, *ACS Appl. Mater. Interfaces* 2013, **5**, 9172–9179). Thus graphene oxide nanosheets provide a large number of anchor sites for the initial monomers and therefore possibly participate in the polymerization. We have carried out additional control experiments to verify this assumption, preparing mixtures of the monomers with graphene oxide to analyze possible interactions by XPS.

In the revised manuscript and Supplementary Information, the following discussion was added:

Pages 4-5, line 106-110 “The formation of COF/rGO composites is possibly enabled by the presence of oxygen containing functional groups on graphene oxide, which promote attractive interaction or even the grafting of Dq molecules by various organic reactions (Supplementary Fig. 6). These reactions might represent the initial step of COF formation on the graphene sheets followed by COF growth after Tp is added.”

Supplementary Figure 6. XPS spectra for the N 1s peak of PTSA-Dq-GO and PTSA-Dq-rGO.

Pages 6-7 of Supplementary Information “To illustrate the possible interaction between graphene oxide and the monomers in the initial polymerization of the COF, mixtures of the amine-functionalized monomer Dq and GO were prepared and analyzed by XPS at different stages, using reaction conditions comparable to the ones used for COF formation. (1) Preparation of PTSA-Dq-GO: Well-ground PTSA (59.4 mg, 0.31 mmol), Dq (13.4 mg, 0.056 mmol) and 5 mL of water were mixed thoroughly and shaken well in a vortex shaker for 5 min. The yellow solution was added into 4.3 mL of 5 mg mL⁻¹ GO dispersion dropwise and stirred for 30 min to obtain a

homogeneous dispersion. After freeze-drying, PTSA-Dq-GO was obtained. The high-resolution N1s spectrum of PTSA-Dq-GO shows two components: mainly protonated amine ($-NH_3^+$, 401.4 eV) and some free amine ($-NH_2$, 399.0 eV) groups^{1,2}. This is expected when adding PTSA to the amine-functionalized Dq monomer, showing that at this stage interactions between GO and Dq are exclusively electrostatic and/or hydrogen bond interactions. (2) The PTSA-Dq-GO mixed solution was then transferred into an autoclave and heated at 120 °C for 24 h. Then, the hydrogel was freeze-dried to obtain PTSA-Dq-rGO. The high-resolution N1s spectrum of PTSA-Dq-rGO reveals again the presence of protonated amine (401.2 eV) but also amide (398.5 eV) and C-NH-C (399.5 eV) groups³, suggesting that covalent grafting of Dq on graphene oxide is possible during the hydrothermal process and thus might be an initial step for growing the COF on the graphene sheet.”

Regarding the second comment: we see a clear peak shift from $\sim 24^\circ$, corresponding to the (002) plane of stacked graphene sheets to $\sim 26^\circ$ in the COF/rGO hybrids (Supplementary Fig. 9) comparable to the signal of the pure COF. This indicates that single graphene sheets are covered with stacked COF layers in the hybrid. Indeed for a simple mixture of rGO and COF, we would expect to see two peaks at 24° and 26° , one for the stacking of graphene layers, the other for stacking of COF layers (or, more probably, as these peaks are not very sharp and intense, just one broader peak covering the angles from 24 - 26° can be expected). To verify this assumption we prepared such a simple physical mixture, by mixing the pure COF and rGO to yield (COF-rGO-mix) and measured the XRD again. As you can clearly see, the (002) peak in this mixture differs from the COF/rGO hybrid and indeed, just one broad peak is seen, which seem to be assembled from both peaks of the layered materials (as rGO exhibits a higher electron density, this peak has the main contribution). In contrast, for the COF/rGO hybrid, we observe a peak shift and no contribution of the (002) peak of rGO, thus we can assume that stacked COF layers have formed on the surface of individual rGO layers. We however admit that from these broad peaks it is hard to make a solid conclusion; therefore we also called this finding a “first hint” in the

manuscript. Since the original description of the PXRD results seems to be a bit confusing, we have revised the related discussion in the text on **Page 5** as:

Page 5 line 125-129 *“It should be however noted, that this peak is shifted to slightly higher angles than observed for pure rGO (Supplementary Fig. 9), giving a first hint that individual graphene layers are covered by the COF, as in a physical 1:1 mixture of both layered materials a broader peak combined from both (002) contributions would be expected.”*

Supplementary Figure 9. PXRD patterns of rGO, COF/rGO and a physical mixture of the pure COF and pure rGO (COF-rGO-mix). The COF-rGO-mix was obtained by mixing and grinding COF and rGO with a mass ratio of 1:1.

2. No evidence in Fig. 1d and 1ec could be found to prove the formation of an additional macroporosity in COF/rGO.

Response: This is correct, but the method applied here, namely nitrogen sorption, does not allow detecting the diameter of pores larger than 40-50 nm. However, the pore sizes of COF/rGO networks are in the range of several micrometers, which are too large to be analyzed using gas sorption. SEM images can provide direct evidence to prove the formation of the macroporosity (Figure 2b).

3. The rGO aerogel with such low BET surface area ($37 \text{ m}^2 \text{ g}^{-1}$) was demonstrated to

show an absorption capacity of 66-93 times, which indicated that these macropores played the vital role for the absorption of organic solvents. As such, why the incorporation of microporous COF could enhance the absorption capacity? And why did the sample of monomers:GO ratio of 1:1 show the highest absorption capacity? Similar concern also exists in their electrochemical properties. Why the insulated COF component could boost the capacitance of rGO aerogel?

Response: Thank you very much for this insightful comment. It is of course correct, that the macropores play the vital role for the high volumetric amount of absorbed solvents. Thus the most important effect of the COF is that it acts as expansion agent, which is largely increasing the porosity of the composites compared to the pure rGO aerogels, as seen from the density measurements. This provides an additional pore volume for the absorption of solvents. However, it should be also noted, that due to their large surface area and pore volume, as well as low density, also microporous COF materials alone possess high absorption capacities for organic solvents, thus the COF provides additional absorption sites. As a result, the formation of the COF/rGO increases the absorption capacity of the aerogels to a certain limit. As we described in the paper, the lowest density is achieved when an optimum ratio for rGO/COF of 1:1 is reached. When the COF loading is further increased, also the density of the aerogels increases again, which has then a detrimental effect on the absorption capacity. Therefore, COF/rGO aerogel prepared from monomers: GO ratio of 1:1 show the highest absorption capacity for a variety of organic solvents.

The related discussion has been added the revised manuscript on **Page 10** as:

Page 10, line 247-254 *“The absorption capacities of the hybrid aerogels are much higher, showing the influence of the overall lower density, higher surface area and porosity. The COF/rGO aerogel prepared from the monomer: GO ratio of 1:1 shows the highest absorption capacity for a variety of organic solvents. This is in line with the decreasing density of the composite compared to pure rGO, with a minimum in density at a 1:1 ratio. In addition, the microporous COF provides a large specific*

surface area, high pore volume and low density, thus itself possess good absorption capacity for organic solvents.”

For the increasing capacitance of the composites, another effect is responsible, namely the additional surface area (increasing double layer capacitance) and redox-active sites (adding pseudocapacitance). Thus the rGO and COF are synergistically affecting the capacitance, the rGO provides conductivity, while the COF adds the surface area and surface redox functionality. Also these effects have an optimum ratio of both components. When the COF amount further increases, the COF coating layer on rGO becomes too thick to exploit the conductivity of rGO. We tried to explain this now in more detail:

Page 12, line 306-314 *“The high specific capacity and rate capability of the COF/rGO electrode is attributed to a synergistic effect of rGO providing conductivity and the COF providing a high surface area and redox active sites, thus increasing the double layer and pseudocapacity, respectively. Furthermore, the formed 3D network is beneficial for rapid charge transfer and ion diffusion to redox-active sites. As a result, the aerogels prepared using a monomer: GO ratio of 1:1 show the highest specific capacitance. Again an optimum COF/rGO ratio is found, as with higher COF amounts the coating layer of the insulating COFs probably becomes too thick, resulting in lower rate capabilities (Supplementary Fig. 29).”*

4. The fabrication method of electrode for pristine COF powder should be added in the experiment part.

Response: We thank reviewer for this constructive suggestion. The related description has been added to the Methods Section on **Page 15**.

Page 15, line 376-380 *“The COF can be directly used as a free-standing electrode by tableting the TpDq-COF powder under a pressure of 5 kN for 2 min (Supplementary Fig. 24). The thickness of the COF electrode is about 75 μ m. The device configuration*

for the COF electrode is the same as for the COF/rGO electrode (Supplementary Fig. 23).”

Supplementary Figure 24. (a) A two-electrode symmetrical supercapacitor assembly prepared with the pure TpDq-COF as electrode. (b) Cross section, and (c, d) surface SEM images of the COF electrode.

Reviewer #2 (Remarks to the Author):

In this manuscript, the authors described an interesting method for the preparation of 3D hierarchical porous covalent organic frameworks (COFs) beyond the traditional powdery COFs. The COFs are grown in situ along the surface of reduced graphene oxide (rGO) to form a hybrid aerogel (COF/rGO), which exhibits impressive absorption capacity towards organic solvents and delivers as well superior electrochemical performance as electrodes of supercapacitors. The strategy demonstrated here enriches the COF-based materials and facilitates the potential applications of COFs. Thus, this paper is acceptable for publication in Nature Communications, however the following issues should be addressed carefully.

1. Page 9, line 235-239: “The pure rGO aerogel without COF shows an absorption capacity of 66–93 times its own weight depending on the organic solvents

(Supplementary Fig. 19a). The absorption capacities of the hybrid aerogels are much higher, showing the influence of the overall lower density, higher surface area and porosity.” The explanation is not so convincing. As we know, graphene aerogels with huge absorption capacity can be obtained as well by simply controlling the architectures, and actually various graphene aerogels have been reported. So, the role of the COF in the COFs/rGO hybrids should be further supported and discussed.

Response: We thank reviewer for this comment. This issue has also been raised by Reviewer 1 (R1.3).

The related discussions has been added into the revised manuscript on **Page 10** as:

Page 10, line 247-254 *“The absorption capacities of the hybrid aerogels are much higher, showing the influence of the overall lower density, higher surface area and porosity. The COF/rGO aerogel prepared from the monomer: GO ratio of 1:1 shows the highest absorption capacity for a variety of organic solvents. This is in line with the decreasing density of the composite compared to pure rGO, with a minimum in density at a 1:1 ratio. In addition, the microporous COF provides a large specific surface area, high pore volume and low density, thus itself possess good absorption capacity for organic solvents.”*

2. COFs/rGO hybrids are described as pseudocapacitive electrode materials, and their electrochemical properties in SCs are evaluated with specific capacitance which maybe inappropriate although many literatures have described their electrode materials exhibiting similar behaviors in the same way. The “pseudocapitance” is assigned to the electrode materials which exhibit a linear dependence of the charge stored with the width of the potential window. While, the specific capacitance of COFs/rGO are “average” values calculated over a limited potential window, which is not constant. As a result, the capacity in $C\ g^{-1}$ or $mAh\ g^{-1}$ should be chosen to evaluate the electrochemical performance instead of the specific capacitance. Detailed information about this issue is available in these literatures (“To Be or Not To Be

Pseudocapacitive?”, Journal of The Electrochemical Society, 162 (5) A5185-A5189 (2015); “Redox-Enhanced Electrochemical Capacitors: Status, Opportunity, and Best Practices for Performance Evaluation”, ACS Energy Lett. 2017, 2, 11, 2581-2590) or others.

Response: We thank reviewer for this constructive suggestion. We have thoroughly read these suggested articles and agree with the reviewer’s opinion that the capacity is more appropriate to evaluate the electrochemical performance. We have added the specific capacity $C \text{ g}^{-1}$ in the revised manuscript. However, for most of the reported electrode materials, the capacitive performance is evaluated by specific capacitance. For comparison with other materials, we therefore also retained the average specific capacitance ($F \text{ g}^{-1}$). In a very recent review article (DOI: 10.1021/acs.chemrev.0c00170), the author also compared specific capacitance and specific capacity at the same time.

The related discussion has been added into the revised manuscript on **Page 11, Page 12** and **Page 15** as:

Page 11, line 274-277 “*Apart from the good conductivity and mechanical strength of graphene, Dq in the COFs backbone can act as an active redox unit to provide the reversible Faradaic reaction in electrochemical energy storage.*”

Page 12, line 298-306 “*In order to get a direct comparison of the capacitive performance, the specific capacitance and capacity is calculated at different current densities from the discharging curve (Fig. 4c, Supplementary Fig. 29 and Supplementary Table 3). Among them, the COF/rGO aerogel yields the highest specific capacitance of $269 F \text{ g}^{-1}$ in a potential window of 1.5 V (equivalent to a specific capacity of $404 C \text{ g}^{-1}$) at the current density of $0.5 A \text{ g}^{-1}$. With the current density increasing to $10 A \text{ g}^{-1}$, the COF/rGO can still deliver a specific capacitance of $222 F \text{ g}^{-1}$ ($292 C \text{ g}^{-1}$) with the retention of 83% capacitance (Fig. 4c).*”

Page 15, line 394-397 “The specific capacity at each electrode was calculated using the equation: $Q = C_g \times \Delta V = 4I\Delta t/m$, where Q ($C g^{-1}$) is the specific capacity stored, I = constant discharge current, Δt = discharge time, m is the total mass of both electrodes.”

Fig. 4 Performance of COF/rGO electrodes in a symmetrical supercapacitor device. **a**, CV curves for rGO, COF/rGO and COF at 50 mV s^{-1} . **b**, The galvanostatic charge-discharge curves of COF/rGO at a current density of 0.5, 1, 2, 3 and 4 A g^{-1} . **c**, The specific capacitances and specific capacities calculated from the discharge curves under different current density. **d**, Comparative bar chart expressing the high performance of COF/rGO among all COF-based supercapacitors in the two-electrode system. **e**, The cyclic stability of COF/rGO at a current density of 8 A g^{-1} . **f**, Impedance spectra of the rGO and COF/rGO capacitor.

Supplementary Table 3. Comparison of specific capacitance and specific capacity for COF/rGO, rGO and COF based electrode materials in this work.

Current density (A g^{-1})	COF/rGO		rGO		COF	
	F g^{-1}	C g^{-1}	F g^{-1}	C g^{-1}	F g^{-1}	C g^{-1}
0.5	269	404	147	219	2.4	1.6
1	258	379	137	201	1.8	1.4
2	245	355	128	187	2.4	1.2

3	239	343	124	180	2.9	0.7
4	237	335	122	176	3.6	0.5
5	232	326	119	170	0	0
10	222	292	113	149	0	0

Supplementary Figure 29. The specific capacitance and specific capacity of COF/rGO hybrids of 0.5:1 and 2:1 calculated from the discharge curves under different current density.

3. Page 11, line 269-271: “The electrochemical capacitance of pure COF is very poor without any charge and discharge capacity due to its insulation property (Supplementary Fig. 22).” The pure COF was then used as the electrodes directly and was compared to the COF/rGO electrode. This comparison is somehow “unfair” because pure COF is insulated and thus does not deliver electrochemical performance. It is maybe more reasonable that pure COF and equivalent conductive materials, such as rGO, should be chosen to fabricate the electrodes and compare to the COF/rGO.

Page 11, line 275-278: “The COF/rGO hybrids display triangular shape with

partial deformation, whose additional capacitance result from the pseudocapacitance effects induced by redox active anthraquinone”. It may not appropriate that the authors ascribe the additional capacitance only to the pseudocapacitance. The specific surface area of COF/rGO ($246 \text{ m}^2 \text{ g}^{-1}$) is higher than that of rGO ($37 \text{ m}^2 \text{ g}^{-1}$). That means COF/rGO should possess higher electric double-layer capacitance compared to rGO. Thus, the additional performance should include pseudocapacity and extra electric double-layer capacity.

Response: We thank reviewer for these constructive suggestions. As the reviewer recommended, we have prepared a mixture of the pure COF with a conductive material, here carbon super P to prepare an electrode, which was then tested under the same conditions for comparison.

The related description has been added on **Page 12, Page 15 of main text and Page 16 of Supplementary Information** as:

Page 12, line 285-290 “*To elucidate if the formation of a composite, i.e. the growing of COF layers on rGO, is indeed necessary or beneficial for the capacitive performance, TpDq-COF was also just physically mixed with a conductive additive namely carbon black (super P). However, the specific capacitance of the COF-carbon black mixture (COF-C-mix) is only 22.4 F g^{-1} (32 C g^{-1}) at 0.5 A g^{-1} although the conductive carbon was introduced (Supplementary Fig. 26).*”

Page 15, line 380-385 “*A mixture of COF and conductive material was obtained by mixing and grinding 58.5 wt.% of COF and 41.5 wt.% of carbon super P. The resulting powder was tableted under a pressure of 5 kN for 2 min to obtain the COF-C-mix electrode. The mass of both electrode materials was $\sim 1.0 \text{ mg}$. The specific capacitance and capacity of COF-C-mix is calculated based on the mass of the COF.*”

Supplementary Figure 26. (a) Cross section, and (b, c) surface SEM images of the COF-C-mix electrode. (d) CV curves and (e) the galvanostatic charge-discharge curves of COF-C-mix. (f) The specific capacitance and specific capacity of COF-C-mix calculated from the discharge curves under different current density.

As the reviewer suggested, we have also corrected the description of the CV curves for COF/rGO hybrids in the revised manuscript on **Page 12** as follows:

Page 12, line 295-298 “The COF/rGO hybrids display triangular shape with partial deformation, whose additional capacity results from the pseudocapacity induced by redox active anthraquinone and extra electric double-layer capacity generated by enhanced specific surface area.”

Reviewer #3 (Remarks to the Author):

In this manuscript, the authors reported the fabrication of COF/reduced graphene oxide (rGO) aerogels, which show good absorption capacity for organic solvents and can be used as active material of supercapacitor devices with high performance. This is an interesting result, as they provide a new way to fabricate COFs into stable 3D architectures that may have practical applications in many areas. I would like to recommend the publication in Nature Communications after addressing the following

concerns:

1. As mentioned in the introduction part, GO was used to prepare many different composites, such as graphene-supported MOFs. The author should discuss more here about the advantages of using COF. In addition, the authors can compare the performance of COF/GO composite with other reported systems in the main text.

Response: We thank the reviewer for the valuable suggestion. The related discussion about the advantages of using COFs for hybrid preparation has been added into the revised manuscript on **Page 3** as:

Page 3, line 61-64 *“COFs exhibit low density, good chemical stability, large surface area and their backbone functionality can be tailored by using appropriate monomers. 2D COFs, which furthermore possess a π -conjugated structure, should be perfectly suited to form composites with 2D graphene.”*

Also, the comparison of the electrochemical performance with other reported systems have been provided in Supplementary Table 5.

Page 13, line 316-317 *“Moreover, it is also comparable with other potential electrode materials (Supplementary Table 5)⁴⁸⁻⁵⁵.”*

2. The author chose TpDq-COF to demonstrate their idea. How about the universality of this method? Is it possible to construct with other COFs systems, such as some highly porous 3D COFs (Angew. Chem. Int. Ed., 2019, 58, 9770)?

Response: We are grateful to the reviewer for this constructive suggestion. To generalize the methodology for 2D COFs, we have tried another amine linker (2,2'-bipyridine-5,5'-diamine (Bpy)) and successfully prepared the corresponding TpBpy-COF using the same procedure as TpDq-COF (Supplementary Fig. 3). The corresponding discussion has been added to the revised Supplementary Information. For the synthesis of porous 3D COFs, the synthesis conditions are usually relatively

harsh, using chloroform, n-butanol, O-DCB as solvents, and highly sensitive to the solvent combinations (*Angew. Chem. Int. Ed.* 2019, **58**, 9770; *Nat Commun* 2018, **9**, 5234). However, we are synthesizing the COF/rGO composites using aqueous GO solution, which may hamper the synthesis of 3D COFs. In these circumstances, for the current project, we have not explored the extension of dimensionality (from 2D COFs to 3D COFs) for the composite preparation. It should be also noted that for the formation of composites, the combination of two 2D materials seems to be more suitable to achieve an intimate combination of the layered compounds.

Supplementary Figure 3. (a) Schematic representation of the synthesis of the TpBpy-COF. (b) PXRD pattern of TpBpy-COF and the comparison with simulated XRD pattern from the modelled structure in eclipsed form. Inset is the space-filling packing model of TpBpy. (c) N₂ adsorption-desorption isotherms and (d) pore size distribution for TpBpy-COF.

Pages 4-5 of Supplementary Information “The hydrothermal method is also applicable to TpBpy-COF (Bpy = 2,2'-bipyridine-5,5'-diamine). Well-ground PTSA

(59.4 mg, 0.31 mmol), Bpy powder (10.4 mg, 0.056 mmol) and 5 mL of water were mixed thoroughly and shaken well in a vortex shaker for 5 min. Then, 7.8 mg of Tp (0.037 mmol) was added into the yellow solution and shaken for another 20 min. The solution was transferred into an autoclave and heated at 120 °C for 24 h. Then, the obtained solid was sequentially washed with hot water, acetone to remove unreacted reagents and monomer fragments. Finally, the material was filtered, collected and oven-dried at 80 °C. The obtained TpBpy-COF display a peak at 3.6° (2θ) corresponding to the reflection from the (100) plane of TpBpy-COF, in good agreement with the corresponding simulated XRD pattern from the modeled structure, confirming the formation of the crystalline structure of TpBpy-COF (Supplementary Fig. 3b). N_2 adsorption isotherm of TpBpy-COF displays a type I isotherm with a steep increase at low relative pressures, indicating the microporosity of TpBpy, which was further demonstrated using the corresponding pore size distribution (Supplementary Fig. 3c-d). TpBpy-COF has a BET surface area of $331\text{ m}^2\text{ g}^{-1}$ and a pore volume of $0.344\text{ cm}^3\text{ g}^{-1}$.”

3. According to the literature (J. Am. Chem. Soc. 2013, 135, 16821), TpDq-COF has a pore size at 2.0 Å. However, Figure 1e indicated TpDq-COF has two different pores. Why? The author can give some explanation here.

Response: The sharp peak in the pore size distribution at ~4 nm can be assigned to the sudden drop in desorption branch around a relative pressure of $P/P_0 = 0.5$, as the condensed N_2 cannot stay in larger pores below this pressure. The defects or larger pores (hysteresis) present in TpDq-COF keep some nitrogen trapped, which is released suddenly at this point, showing a second size of pores in that region. A comparison to the publication mentioned is unfortunately not possible, as the graph for the pore size distribution in this publication was cut off at 3 nm.

4. The authors claimed the TpDq-COF grow uniformly along the surface. Please give some direct evidences here.

Response: The STEM-EDS elemental mapping of COF/rGO (Supplementary Fig.

14c) shows a uniform distribution of nitrogen, which is the first evidence for a uniform loading of the TpDq-COF on the surface of rGO. From AFM images and the corresponding height profiles, the increase of average thickness from 1.5–2.0 nm for pure rGO sheets to 2.9–6.0 nm for COF/rGO sheets further demonstrates that the graphene nanosheets were fully and evenly covered by TpDq-COFs (Fig. 2e, f and Supplementary Figs. 15, 16). In addition, no isolated COFs particles were detected on the graphene nanosheets, indirectly indicating that the COF grow uniformly along the surface of graphene (Fig. 2b, c). We hope that these are enough direct evidences for the COF growth on the graphene sheets.

5. Some related literatures about COFs with new applications (Nat. Commun., 2018, 9, 5234; Nat. Commun., 2019, 10, 2467; etc) can be added in the revised manuscript.

Response: We have cited these two articles about new applications of COFs and added the corresponding summary in the introduction section in the revised manuscript:

Line 30-34 on page 2 *“Because of their structural diversity, permanent porosity, long-range order and the versatile functionalities, which can be introduced into the organic backbone, COFs are promising materials for a range of applications such as organocatalysis⁵, gas storage⁶, molecular separations⁷⁻⁸, energy storage⁹, photocatalytic water splitting¹⁰, light-emitting diodes¹¹, etc.”*

The above-mentioned references have been cited as Refs. 10 and 11 in the revised manuscript.

(10) Ding, H. et al. An AIEgen-based 3D covalent organic framework for white light-emitting diodes. *Nat Commun* **9**, 5234 (2018).

(11) Bi, S. et al. Two-dimensional semiconducting covalent organic frameworks via condensation at arylmethyl carbon atoms. *Nat Commun* **10**, 2467 (2019).

REVIEWERS' COMMENTS:

Reviewer #1 (Remarks to the Author):

The authors have sufficiently addressed all my questions/comments. The revised manuscript can be accepted for publication in Nature Communications.

Reviewer #2 (Remarks to the Author):

The authors have carefully revised the manuscript by addressing all the comments and suggestions, and the quality of the Ms is significantly improved, which is acceptable for publication in NC.

Reviewer #3 (Remarks to the Author):

The authors have answered the problems I raised well. I am satisfied with the changes and recommend the publication.

Response to reviewers' comments

Reviewer #1 (Remarks to the Author):

The authors have sufficiently addressed all my questions/comments. The revised manuscript can be accepted for publication in Nature Communications.

Response: We thank the reviewer for recommending publication of our manuscript in Nature Communications.

Reviewer #2 (Remarks to the Author):

The authors have carefully revised the manuscript by addressing all the comments and suggestions, and the quality of the Ms is significantly improved, which is acceptable for publication in NC.

Response: We thank the reviewer for recommending publication of our manuscript in Nature Communications.

Reviewer #3 (Remarks to the Author):

The authors have answered the problems I raised well. I am satisfied with the changes and recommend the publication.

Response: We thank the reviewer for recommending publication of our manuscript in Nature Communications.